# Socioeconomic determinants of cancer screening utilisation in Latin America: A systematic review

Bernardo Nuche-Berenguer[1], Dikaios Sakellariou[2]*

1 Former student, London School of Hygiene and Tropical Medicine, London, United Kingdom, 2 School of Healthcare Sciences, Cardiff University, Cardiff, United Kingdom

* sakellarioud@cardiff.ac.uk

## Abstract

### Introduction

Cancer incidence and mortality in Latin America are rising. While effective cancer screening services, accessible to the whole population and enabling early cancer detection are needed, existing research shows the existence of disparities in screening uptake in the region.

### Objective

We conducted a systematic review to investigate the socioeconomic determinants for the disparities in the use of breast, cervical and colorectal cancer screening services in Latin America.

### Methods

We searched for studies reporting on socioeconomic determinants impacting on access to breast, cervical and colorectal cancer screening, published from 2009 through 2018. The studies that qualified for inclusion contained original analyses on utilisation of breast, cervical and colorectal cancer screening across socioeconomic levels in Latin America. For each study, paired reviewers performed a quality analysis followed by detailed review and data extraction.

### Results

Twenty-four articles that met the eligibility criteria and were of sufficient quality were included in this review. Thirteen of the included articles were written in English, eight in Portuguese and three in Spanish, and they reported on the use of breast or cervical cancer screening. No studies were found on the socioeconomic determinants regarding the utilisation of colorectal cancer screening in Latin America. Low income, low education level, lack of health insurance and single marital status were all found to be determinants of underuse of breast and cervical cancer screening services.

**Data Availability Statement:** All relevant data are within the manuscript and its Supporting Information files

**Funding:** The authors received no specific funding for this work.

**Competing interests:** The authors have declared that no competing interests exist.

## Conclusions

Cancer screening programs in the region must prioritize reaching those populations that underuse cancer screening services to ensure equitable access to preventive services. It is important to develop national screening programmes that are accessible to all (including uninsured people) through, for example, the use of mobile units for mammography and self-screening methods.

## Introduction

In Latin America (LA), there are approximately 1.4 million new cases and 670,000 deaths from cancer each year. [1] It is estimated that by 2040 there will be a 78% increase in cancer incidence and a 93% increase in cancer mortality in the region. [1] Although primary prevention is important for mitigating the burden of cancer, investing in cancer management, encompassing screening, detection, treatment and palliative care, is essential in reducing mortality rates. [2] Since cancer in LA is often diagnosed in its late stages,[3] establishing cancer screening services and encouraging participation in screening programs is crucial to reducing cancer mortality. However, not all cancer types are eligible for screening and the currently available evidence supports the effectiveness of screening for breast,[4, 5] cervical,[6] and colorectal cancer[7, 8] in preventing cancer deaths. Breast and cervical cancer screening are recommended by the World Health Organization (WHO), and colorectal cancer screening is recommended by other organizations, such as the American Cancer Society.

In line with these international recommendations, most countries in LA have established national screening programs for breast and cervical cancer and, to a lesser extent, colorectal cancer.[9] However, these programs face challenges in reaching all geographical areas and may not be equally accessible across all socioeconomic strata.[10, 11] The relationship between socioeconomic status and access to health services has been well documented.[12] Some U.S. studies have explored the association of socioeconomic factors with the uptake of cancer screening services,[13] but there is limited evidence from LA, a region with high inequalities in socioeconomic indicators. Identifying information on socioeconomic determinants of cancer screening underuse is a key factor in the design of strategies that encourage and enable the use of these services and in reducing the disparity in cancer outcomes in LA. We conducted a systematic literature review with the aim to identify the socioeconomic determinants that affect utilisation of cancer screening services in LA.[14] The research question, framed according to the PICOS framework, was: "What are the socioeconomic characteristics that impact on access to cancer screening services in Latin America?" The results of this study will be useful in guiding the countries in the region develop policies aimed at reducing disparities in cancer screening utilisation.

## Methods

### Search strategy

The review took place between May and July 2018. No publicly available protocol exists (see S1 File for the PRISMA checklist). We identified three key search concepts and their synonyms (see Table 1).

We searched the following databases: MedLine (via Ovid), Embase (via OVID), LILACs (via BIREME. The Latin American and Caribbean Center on Health Sciences), and Global

**Table 1. Search concepts identified from the research question.**

| Socioeconomic Factors | Cancer screening services | Latin America |
|---|---|---|
| social class | cancer prevent* services | Argentina |
| socio-economic class | cancer screen* | Bolivia |
| socio-economic level | breast cancer screen* | Brazil |
| strat* | cervical cancer screen* | Chile |
| social status | colorectal cancer screen* | Colombia |
| economic status | mammography | Costa Rica |
| educational status | Pap Smears* | Cuba |
| education* level | Human Papillomavirus (HPV) test* | Dominican Republic |
| profession* class | visual inspection with acetic acid | Ecuador |
| professional level | VIA | El Salvador |
| | colonoscopy | Guatemala |
| | f?ecal occult blood test | Haiti |
| | Papanicolaou | Honduras |
| | | Mexico |
| | | Nicaragua |
| | | Panama |
| | | Paraguay |
| | | Peru |
| | | Puerto Rico |
| | | Uruguay |
| | | Venezuela |

Health (via OVID). These databases were identified as the most comprehensive for the topic of participation in cancer screening services. Moreover, LILACS is the most important and comprehensive index of scientific and technical literature for LA.

The three search concepts, their synonyms (using truncations and wildcards- indicated in Tables 1 and 2 by the use of the asterisk '*'- where appropriate), and subject heading terms

**Table 2. Combination of search terms used in the final search strategy for the literature review.**

| | Combination of Search Terms |
|---|---|
| 1 | Socioeconomic Factors OR social class OR socio-economic class OR socio-economic level OR social strat* OR social status OR economic status OR educational status OR education* level OR profession* class OR professional level |
| 2 | exp socioeconomic factors/ (only socioeconomics in Global Health) |
| 3 | 1 OR 2 |
| 4 | Cancer prevent* services OR cancer screen* OR breast cancer screen* OR cervical cancer screen* OR colorectal cancer screen* OR mammography* OR Pap smears OR Papanicolaou* OR visual inspection with acetic acid OR VIA OR HPV test* OR colonoscopy OR F?ecal occult blood test |
| 5 | exp cancer screening (preventive services did not exist) |
| 6 | 4 OR 5 |
| 7 | Latin America OR Argentina OR Bolivia OR Brazil OR Chile OR Colombia OR Costa Rica OR Cuba OR Dominican Republic OR Ecuador OR El Salvador OR French Guiana OR Guatemala OR Haiti OR Honduras OR Mexico OR Nicaragua OR Panama OR Paraguay OR Peru OR Puerto Rico OR Uruguay OR Venezuela |
| 8 | exp Latin America/ |
| 9 | 7 OR 8 |
| 10 | 3 AND 6 AND 9 |
| 11 | Only articles published from 2009 until 2018 in English, Portuguese or Spanish |

were combined, using the Boolean operators 'OR', within concepts, and 'AND' to combine concepts to develop the final search strategy (see Table 2 for search strategy). In order to conduct the search in LILACS, where a large proportion of articles are written in Portuguese or Spanish, the same search terms were used in English as most abstracts have been translated.

## Inclusion and exclusion criteria

Only articles reporting on breast, cervical and colorectal cancer screening were included because these are the screening types consistently supported by evidence[4–8] and that are recommended by international health organizations. The review included articles reporting on studies from LA, written in English, Spanish or Portuguese and published in peer-reviewed journals since 2009, because we were looking for updated information as the implementation of new cancer norms and screening programs in Latin America is expected to have an impact in the access to cancer screening services. The inclusion criteria are outlined in Table 3 and are presented according to the PICOS format. The English-speaking Caribbean region was not included in this review because it is considered as a different sociopolitical region with different geographical and cultural characteristics.

## Data extraction

The studies retrieved through the search process were assessed for inclusion using pre-determined selection criteria based on the research question. First, the title and abstract were scanned to check whether the study:

1. Referred to or was based on participation in cancer screening.

2. Assessed the influence of socioeconomic indicators on cancer screening uptake.

3. Was conducted in Latin America.

All three conditions needed to be present for the study to be included. When it was not clear whether these criteria had been met, the full article was examined. 127 articles were subjected to full review (74 in English, 36 in Portuguese and 17 in Spanish).

To ensure consistency in the study selection process, the following definitions were used:

- **Cancer screening services:** healthcare services aimed at detecting breast, cervical or colorectal cancer in asymptomatic populations.

**Table 3. Inclusion criteria.**

| PICOS Element | PICOS question term | Variables considered |
|---|---|---|
| Population | General population in Latin America. | Population from the 21 countries specified in Table 1. |
| Intervention | Cancer screening. | Breast, cervical, and colorectal cancer screening. |
| Comparison group | Population stratified by socioeconomic determinants. | Income, education, marital status, insurance status, use of other health services. |
| Outcome | Access to cancer screening. | No previous participation in breast cervical or colorectal cancer screening. No breast, cervical or colorectal cancer screening in the previous 1–3 years. |
| Study design | Cross-sectional, cohort, case-control, and randomized studies. | |

- **Socioeconomic level:** this is the combined measure of an individual's or family's economic and social position in relation to others, based on income, education and occupation.

- **Latin America:** only studies conducted in the 21 countries listed in Table 1 were considered.

After reading each article that met the inclusion criteria, a short paragraph summarizing the aim, methods, main findings, strengths and weaknesses of the study was written. This information was entered into a data extraction table (see S1 Table).

### Quality assessment

The quality of the studies was assessed using a modified version of the U.S National Institutes of Health (NIH) Quality Assessment Tool for Observational, Cohort and Cross-Sectional Studies (https://www.nhlbi.nih.gov/health-topics/study-quality-assessment-tools). The definitions of "high", "satisfactory" and "poor" quality that were used were based on the ten questions in the NIH Quality Assessment checklist that were deemed relevant for cross-sectional studies (see S2 Table), while questions not relevant for cross-sectional studies were removed from the tool. The quality of the studies that scored 8 or more out of 10 criteria was deemed as high; studies that scored 5 to 7 out of 10 criteria were deemed as satisfactory; finally, studies that scored 4 or less out of 10 criteria were considered of poor quality and were excluded from the review.

## Results

Fig 1 shows the selection process of the identified articles. A total of 840 records were initially identified. A title or title and abstract screening of the initial 840 articles narrowed these down to 127 articles for full-text review; 26 articles met all eligibility criteria. Of these articles, 15 were from Brazil, 2 from Colombia and 1 each from Argentina, Bolivia, Chile, Costa Rica, Dominican Republic, Haiti, Honduras, Mexico, Peru and Puerto Rico. No articles were found that assessed socioeconomic determinants on the use of colorectal cancer screening programs in LA.

### Quality

The 26 identified studies were cross-sectional and were critically appraised by applying a modified version of the NIH quality assessment tool. Of the 26 studies, 11 scored 8–10 (out of the 10 NIH quality criteria) and were designated of high quality; 13 scored 5–7 and 2 studies scored less than 4,[14, 15] and were excluded from the review; therefore, 24 articles (13 in English, 8 in Portuguese and 3 in Spanish) were included in this review (see Fig 1).

### Socioeconomic determinants and access to cancer screening in Latin America

The answer to the study question "What are the socioeconomic characteristics that impact on access to cancer screening services in Latin America?" is summarized in Tables 4 and 5. These tables show in detail the association between socioeconomic determinants and Pap test (Table 4) and mammography (Table 5) utilisation. Below we summarize the main aspects found in the included studies.

The outcome measures found across the 24 studies were the self-reported participation in breast or cervical cancer screening. We did not identify any studies on the use of colorectal cancer screening. Within these two outcome measures, two levels of underuse were identified.

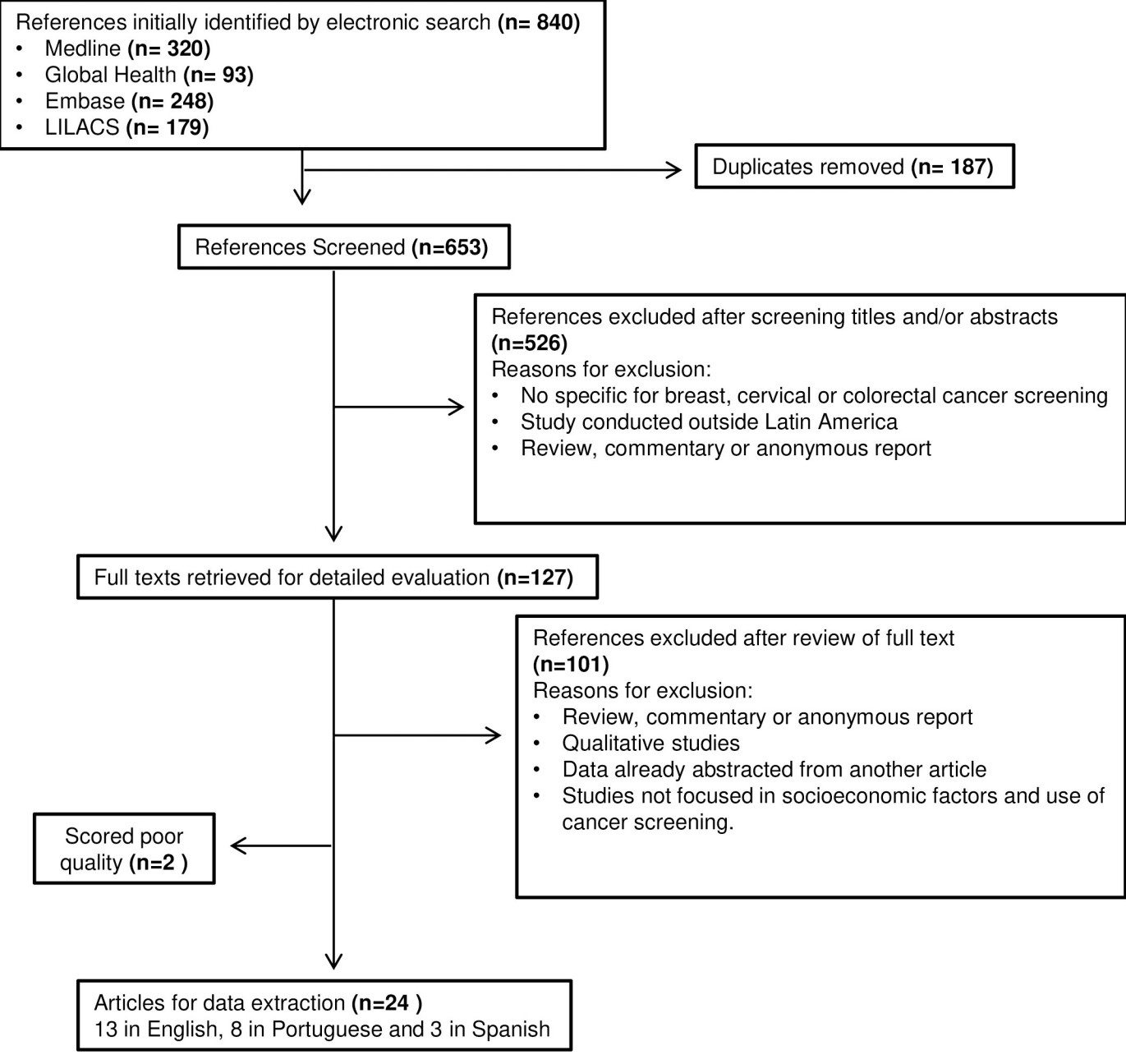

**Fig 1. PRISMA flow diagram of the study selection procedure.**

Firstly, no previous participation in breast or cervical cancer screening. Secondly, no breast or cervical cancer screening in the previous 1–3 years (depending on the study).

The following exposure measures (independent variables) were identified across the included studies: income, education, marital status, health insurance, use of other health services, or other factors, such as smoking status, or number of children. To define the relationship of these variables with participation in cancer screening, only data coming from multivariate analyses adjusted for confounders were considered.

**Table 4. Studies analysing the association between socioeconomic characteristics and cervical cancer screening utilization.**

| Author, Quality score | Setting | Independent variables* | Population of Interest (n) | Income gradients for undergoing Pap | Education gradients for undergoing Pap |
|---|---|---|---|---|---|
| *Albuquerque KM, 2009* [37] **7/10** | Pernambuco, Brazil. | Marital status, no children, education. | Women 18–69. (n = 258) | Not analysed | **Never [OR]** Education (years) **Complete primary or more:** 1 **Incomplete middle-school:** 2.01 (0.70–6.14) **Complete middle-school or more:** 2.97 (1.13–7.82) |
| *Brenes-Camacho,2009* [24] **9/10** | Costa Rica | Education, income, health insurance. | Women over 60 (n = 1464) | | **In previous year [OR]** **Education (years)** **0–3:** 1 **>7:** 0.67* |
| *Ortiz AP, 2010* [25] **5/10** | Puerto Rico | Income, marital status, use of other health services | Women over 18 (n = 2,206) | **In previous 3 years [OR]** **Income (USD)** **<$15,000:** 1 **$15,000–34,999:** 1.29 (1.28–1.31) **$35,000–49,999:** 2.78 (2.71–2.84) **≥$50,000:** 2.45 (2.39–2.50) | |
| *Gasperin SI, 2011* [16] **9/10** | Florianopolis, Brazil | Income, education, marital status, income, age, use of other health services. | Women 20–59 (n = 952) | **No in previous 3 years [OR]** **Income (terciles)** **1st:** 1 **2nd:** 1.07 (1.01–1.13)* **3rd:** 1.06 (1.00–1.11)* **Never [OR]** **Income (terciles)** **1st:** 1 **2nd:** 0.60 (0.40–0.89)* **3rd:** 0.78 (0.45–1.33) | **No in previous 3 years [OR]** **Education (years)** **>12:** 1 **9–11:** 0.93 (0.90–0.96)* **5–8:** 0.93 (0.88–0.98)* **<3:** 0.93 (0.87–0.98)* **Never [OR]** **Education (years)** **>12:** 1 **9–11:** 1.98 (1.28–3.08)* **5–8:** 2.54 (1.53–4.21)* **<3:** 4.74 (2.70–8.31)* |
| *Price J, 2011* [38] **5/10** | Yamaranguila, Honduras. | Distance to the health centers. | Indigenous women over 18 (n = 134) | | Not detailed. |
| *Brischilliari SCR, 2011* [29] **8/10** | Maringa, Brazil. | Income, probably education, occupation. | Women 45–69 (n = 456) | **No in previous 3 years [PR]** **Income (category)** **A/B (richer):** 1 **C/D:** 2.19 (1.17–4.11)* | |

(*Continued*)

**Table 4.** (*Continued*)

| Author, Quality score | Setting | Independent variables* | Population of Interest (n) | Income gradients for undergoing Pap | Education gradients for undergoing Pap |
|---|---|---|---|---|---|
| *Cesar JA, 2012* [17] **8/10** | Rio Grande, Brazil. | Education, age, marital status, unplanned pregnancy, use of other health services. | Pregnant women (n = 2,288) | **Income (quintiles)** $5^{th}$ *(richer)*: 1 $4^{th}$: 1.42 (1.14–1.76)* | **No during pregnancy [PR]** **Education (years)** >**12**: 1 **9–11**: 1.22 (0.93–1.61)* **5–8**: 1.49 (1.12–1.98)* <**4**: 1.54 (1.12–2.12)* |
| *Correa MDA, 2012* [30] **6/10** | Brazil. | Income, education, first pregnancy. | Women that recently gave birth (n = 3,939) | **Income (category)** *B/C (richer)*: 1 *D*: 0.98 (0.89–1.08) *E*: 0.86 (0.78–0.96)" | **In previous 3 years [PR]** **Education (years)** **0–5**: 1 **6–8**: 1.14 (1.03–1.26)* >**9**: 1.26 (1.13 1.39)* |
| *de Maio FG, 2012* [33] **8/10** | Argentina | Income, education | Women over 18 (n = 7620 mammography); (n = 19704, PAP) | **No in previous 2 years [OR]** **Income (category)** *High*: 1 *Medium*: 1.90 (1.59–2.26) *Low*: 3.12 (2.64–3.68) | **Education (level)** *High*: 1 *Medium*: 1.88 (1.73–2.04)* *Low*: 2.91 (2.33–3.63)* |
| *Senicato, 2012* [39] **8/10** | Campinas, Brazil | Education | Women 20–59 (n = 508) | | **In previous 2 years [PR]** No differences. |
| *Barrionuevo-Rosas, 2013* [21] **9/10** | Peru | Education, income, health insurance, place of residence (urban vs rural) | Women 30–49 (n = 12,272) | **Income (level)** *Low*: 1 *Medium*: 1.21 (1.14–1.29)* *High*: 1.25 (1.17–1.33)* | **In previous 5 years [PR]** **Education (level)** *No education*: 1 *Secondary*: 1.10 (1.06–1.15)* *Superior*: 1.21 (1.15–1.26)* |
| *Martinez Mesa J, 2013* [18] **7/10** | Brazil | Education, income, age, race, parity, place of residence, health insurance, use of other health services. | Women 25–64 (n = 102,108) | **Never [PR]** **Income (Quintiles)** $1^{st}$ *(Richer)*: 1 $2^{nd}$: 1.32 (1.16;1.51)* $3^{rd}$: 1.57 (1.38;1.79)* $4^{th}$: 1.88 (1.66;2.13)* $5^{th}$ *(poorer)*: 2.19 (1.91;2.50)* | **Never [PR]** **Education (years)** >**12**: 1 **9–11**: 1.25 (1.15;1.35)* **5–8**: 1.50 (1.40;1.61)* **0–4**: 2.28 (2.11;2.45)* |

(*Continued*)

**Table 4.** (Continued)

| Author, Quality score | Setting | Independent variables* | Population of Interest (n) | Income gradients for undergoing Pap | Education gradients for undergoing Pap |
|---|---|---|---|---|---|
| *Soneji S, 2013 [22]* **5/10** | Brazil, Bolivia, Dominican Rep., Ecuador, Nicaragua and Peru | Education, income, age, place of residence (urban vs rural), use of other health services. | Women over 18. (n = 12,789 Bolivia) n = 20,410, (Dominican Republic) | **Recently [OR] (Bolivia)** **Income (level)** *Poorest*: 0.60 (0.56–0.66)* *Poorer*: 0.79 (0.74–0.84)* *Middle*: 1 *Richer*: 1.12 (1.07–1.18)* *Richest*: 1.29 (1.24–1.35)* **Recently [OR] (Dominican Republic)** **Education (level)** *None*: 0.81 (0.77–0.85)* *Primary*: 1 *Secondary*: 1.04 (1.02–1.06)* *Higher*: 1.02 (0.98–1.05) | **Recently [OR] (Bolivia)** **Education (level)** *None*: 0.79 (0.83–0.85)* *Primary*: 1 *Secondary*: 1.09 (1.06–1.13)* *Higher*: 1.03 (0.99–1.07)* **Recently [OR] (Dominican Republic)** **Income (level)** Poorest: 0.77 (0.74–0.80)* Poorer: 0.91 (0.88–0.94)* Middle: 1 Richer: 1.10 (1.06–1.13)* Richest: 1.20 (1.17–1.24)* |
| *Oliveira MV, 2014 [26]* **5/10** | Vitoria da Conquista, Brazil | Education, income, age, marital status, use of other health services. | Indigenous women age 18 to 64 (n = 348) | **No in previous 3 years [OR]** **Income** *<1/2 MMW*: 1 *>1/2 MMW*: 0.76 (0.37–1.55) **Never [OR]** **Income** *<1/2 MMW*: 1 *>1/2 MMW*: 1.01 (0.58–1.77) | **No in previous 3 years [OR]** **Education (years)** *0*: 1.32 (0.54–3.23) *1–4*: 2.34 (0.89–6.16) *>4*: 1 **Never [OR]** **Education (years)** *0*: 3.26 (1.14–9.30) *1–4*: 2.15 (0.89–5.14) *>5*: 1 |
| *Bermedo-Carrasco S, 2015 [20]* **9/10** | Colombia | Income, education, parity, health insurance, place of residence (urban vs rural), region. | Women over 18 (n = 40,392) | **Income (quintiles)** *1st*: 0.60 (0.52–0.69) *2nd*: 0.80 (0.70–0.91) *3rd*: 0.87 (0.77–0.99) *4th*: 0.94 (0.82–1.07) *5th (richest)*: 1 | **At least once [OR]** **Education (years)** *9–11*: 0.78 (0.65–0.93)* *5–8*: 1.69 (1.54–1.86)* *1–4*: 1.03 (0.69–1.12) *0*: 1 |
| *Sakellariou D, 2017 [36]* **7/10** | Chile | Education, income, health insurance, use of other health services, marital status, occupation. | Disabled women (n = 5,823) |  | **Yes [OR]** *Education (years)*: 1.05 (1.04–1.07) |

**OR**: adjusted odds ratio; **PR**: adjusted prevalence ratio; **Pap**: Pap smear; **MMW**: minimum monthly wage; **USD**: U.S Dollars. *P<0.05

**Income.** Income was estimated in different ways across the included studies. Most studies measured income as a categorical variable measuring in quintiles [16–20] or other categories. [21–25] Other studies used minimal monthly wages [26, 27] or the socioeconomic classification (A to E) established by the Brazilian Institute of Geography and Statistics (https://ww2.ibge.gov.br/home/estatistica/populacao/condicaodevida/indicadoresminimos/defaulttab.shtm).[28–30] A study from Colombia used a wealth index,[31] a well-established indirect

**Table 5. Studies analysing the association between socioeconomic characteristics and mammography utilization.**

| Author, Quality score | Setting | Independent variables* | Population of Interest (n) | Income gradients for undergoing MMG | Education gradients for undergoing MMG |
|---|---|---|---|---|---|
| *Brenes-Camacho,2009* [24] **9/10** | Costa Rica | Education, income, health insurance. | Women over 60 (n = 1464) | **In previous 3 years [OR]** **Income (USD)** *<$100*: 1 *$400*:1.47* | **In previous year [OR]** **Education (years)** *0–3*: 1 *4–6*: 0.63*ᵃ |
| *Novaes CDE, 2009* [35] **8/10** | Minas Gerais, Brazil. | Education, age, marital status, use of other health services. | Women over 60 (n = 4,621) | | **In previous year [OR]** **Education (years)** *>4*: 1 *<4*: 1.1 (1.0–1.1) |
| *Matos JC, 2011* [28] **7/10** | Maringa, Brazil. | Income, education, ethnicity, religion, use of other health services. | Women 40–69 (n = 439) | **At least once [OR]** **Income (category)** *A/B (richer)*: 1 *C*: 0.45 (0.27–0.76) *D/E*: 0.50 (0.13–1.90) | |
| *de Maio FG, 2012* [33] **8/10** | Argentina | Income, education | Women over 18 (n = 7620 mammography); (n = 19704, PAP) | **No in previous 2 years [OR]** **Income (category)** *High*: 1 *Medium*: 1.54 (1.16–2.05) *Low*: 2.37 (1.81–3.11)* | **Education (level)** *High*: 1 *Medium*: 2.30 (1.97–2.69) *Low*: 2.96 (2.33–3.76)* |
| *Lages RB, 2012* [40] **7/10** | Teresina, Brazil | Education, income, race, marital status, smoking, health insurance | Women 40–69 (n = 433) | No significant differences related to income after adjusted analysis. | No significant differences related to education after adjusted analysis. |
| *Senicato, 2012* [39] **8/10** | Campinas, Brazil | Education | Women 20–59 (n = 508) | | **In previous 2 years [PR]** **Education (years)** *>9*: 1 *<9*: 0.63 (0.50–0.80) |
| *Agudelo BM, 2013* [34] **5/10** | Mexico | Education, income, health insurance, place of residence (urban vs rural) | Women 40–59 (n = 12,281) | Not detailed | Not detailed |
| *Font-Gonzalez, 2013* [31] **8/10** | Colombia | Education, income, health insurance, ethnicity, marital status use of other health services. | Women 40–69 (n = 27,116) | **In previous 2 years [OR]** **Wealth index** *Lowest*: 1 *Medium*: 1.9 (1.5–2.3)* *High*: 2.5 (2.1–3.0)* *Highest*: 4.7 (3.9–5.8)* | **Education (level)** *None*: 1 *Elementary*: 1.3 (1.0–1.6) *Secondary*: 1.6 (1.3–1.9)* *University*: 2.3 (1.8–2.9)* |
| *Schneider IJ, 2014* [19] **7/10** | Florianopolis, Brazil. | Education, income, marital status | Women 40–69 | **Income (quartiles)** *4ᵗʰ*: 1.98 (1.29–3.04) *3ʳᵈ*: 1.51 (0.99–2.30) *2ⁿᵈ*: 1.29 (0.77–2.16) *1ˢᵗ(poorest)*: 1 | **In the previous year [PR]** **Education (years)** *>12*: 4.18 (1.30–13.44)* *9–11*: 4.36 (1.32–14.47)* *5–8*: 3.83 (1.16–12.63)* *1–4*: 3.22 (0.93–11.18)* *0*: 1 |
| *Melo ECP, 2016* [27] **8/10** | Brazil | Income, education, health insurance, race. | Women over 40 | **At least once [OR]** **Per capita family income** *<1/4MMW*: 1 *1/4-1/2MMW*: 1.19 *1/2-1MMW*: 1.56* *1-2MMW*: 1.91* *2-3MMW*: 2.76* *>3MMW*: 4.10* | **At least once [OR]** **Education (years)** *<1*: 1 *1–3*: 1.34* *4–7*: 1.51* *8–10*: 1.86* *11–14*: 1.97* *>15*: 2.33* |
| *De Andrade Souza, 2017* [23] **6/10** | Boa Vista, Brazil | Education, income, use of other health services. | Women 40–69 (n = 240) | **No in previous 2 years [OR]** **Per capita family income** *Low*: 1.33 (0.55–3.22) *Middle*: 1 *High*: 0.49 (0.18–1.35) | **No in previous 2 years[OR]** **Education (level)** *University*: 0.32 (0.17–0.61)* *High school*: 0.96 (0.39–2.34) *Primary*: 1.98 (1.48–3.05)* |

*(Continued)*

**Table 5.** (Continued)

| Author, Quality score | Setting | Independent variables* | Population of Interest (n) | Income gradients for undergoing MMG | Education gradients for undergoing MMG |
|---|---|---|---|---|---|
| *Sakellariou D, 2017 [36]* **7/10** | Chile | Education, income, health insurance, use of other health services, marital status, occupation. | Disabled women (n = 5,731) | | **Yes [OR]** *Education (years):* 1.02 (1.04–1.05) |

**OR:** adjusted odds ratio; **PR:** adjusted prevalence ratio; **MMG:** mammography, **MMW:** minimum monthly wage; **USD:** U.S Dollars.
*P<0.05

measure of income.[32] Several studies found a positive association between income and participation in cervical cancer screening in the previous 1–3 years.[21, 22, 24, 25, 29–31, 33] However, other studies did not find a clear association.[16, 17] When the variable was considered as "PAP smear never used", the association between income and non-participation was also evident.[16, 18, 20]

Studies also reported a positive association between self-reported participation in mammography in the previous 1–3 years and income.[19, 31, 33, 34] However, some studies failed to demonstrate this association.[24] When the variable considered was "mammography never used" an association with income was also consistently found.[27, 28]

**Education level.**   Most studies defined education level as number of years of schooling. [14, 16–20, 24, 26, 27, 30, 35, 36] A few studies defined the level of education as the completion of primary, secondary or tertiary education,[21–23, 33] which makes regional comparisons more difficult.

Numerous studies showed that education level is positively associated with participation in cervical cancer screening both in the previous 1–3 years [14, 16, 17, 21, 30, 33] and also when considering the "PAP smear never used" in cervical cancer screening.[18, 36, 37] However, an important number of studies did not find any association between education level and cervical cancer screening.[25, 38, 39] Another two studies found a possible, but not fully proven, association [26, 29]. Most studies found an association between education levels and participation in mammography in the previous 1–3 years.[19, 23, 31, 34–36, 39] Less clear associations were found in a study in Brazil [23] that only found primary education to be a determinant for participating in mammography; in another study in Costa Rica,[24] education negatively correlated with yearly participation in mammography, although not when the interval was set at three years. This study concluded that better-educated women tend to wait longer between mammography and Pap smear screenings than their less educated counterparts. Another study,[28] did not find any association between mammography screening and levels of education. In one study in Brazil,[27] education was positively associated with never participating in mammography. However, this association was absent in another study.[40]

**Marital status.**   Marital status was consistently analysed as a determinant for participating in cancer screening. Most studies established that single women were less likely to participate in cancer screening,[16, 17, 19, 26, 35–37, 40] with only one study finding a positive association between being single and screening uptake.[14, 25]

**Health insurance.**   Being covered by health insurance was commonly established as a major determinant favouring participation in cancer screening.[18–21, 24, 27, 31, 33, 34, 36, 40] However, some studies found no such association.[23, 25]

**Place of residence.**   A number of studies identified living in a rural area as a barrier for participating in cancer screening [18, 20–22, 34, 36]. In some other countries, such as the Dominican Republic [22], the place of residence was not found to affect participation in cancer

screening, while the analysis of a health survey in Bolivia concluded that women living in rural areas were more likely to have participated in cervical cancer screening. [22]

**Use of other health services.** The use of other health services can be a determinant for encouraging participation in screening, especially in a region where most cancer screening programmes are opportunistic.[3] We identified several studies establishing an association between use of other health services (e.g. maternity care) and higher participation in screening programs.[14, 16–18, 22, 23, 25, 26, 28, 31, 35, 36]

**Other factors.** This literature review identified other factors linked to poor screening performance in LA, including smoking;[14, 17, 40] lack of physical activity;[25] use of oral contraceptives;[15, 17] ethnicity/race (with white populations having lower rates of underuse);[18, 19, 21, 27, 31, 36, 40]; and greater number of children.[18, 21]

## Discussion

Our aim was to investigate the socioeconomic determinants for the disparities in the use of cancer screening services in LA. Twenty-four articles published between 2009 and 2018 were included in this review. Low income, low education level, lack of health insurance and single marital status were all found to be determinants of underuse of breast and cervical cancer screening services in LA. No evidence was found about colorectal cancer screening and this may signal a lack of knowledge that should be considered in future research. With more than 1.4 million new cases and 670,000 deaths every year,[1] cancer is a major public health challenge in LA. Breast, cervical and colorectal cancer are among the most common cancer types [41–42] These numbers are projected to keep growing in the coming years and, therefore, health systems need to provide a comprehensive approach that addresses prevention, screening, early detection and treatment of cancer. Although most countries in LA have introduced screening programmes for breast, cervical and colorectal cancer, [9] these programmes are often opportunistic, not equally accessible to everyone and often lack the required quality.[3] As a result, cancer is still diagnosed late and generally the screening programmes do not have the desired impact of reducing mortality.[42–44] Because LA lacks comprehensive cancer registries, there is no information about the socioeconomic characteristics of cancer patients. However, some studies have concluded that the late diagnosis and mortality from cervical [45–47] and breast [48–52] cancer in the region is more common in lower socioeconomic strata. Although several reasons may explain these differences, it is likely that differential access to preventive and therapeutic healthcare is an important factor.

The results of this literature review are consistent with the existing evidence on the impact of socioeconomic status on accessing healthcare [12] and cancer screening in both high,[13, 53–55] and low and middle-income countries.[56–59] Indeed, two similar systematic reviews previously conducted in the U.S [53] and in low and middle-income countries [57] obtained similar results to this LA-based literature review. Other socioeconomic factors that affect cancer screening uptake include employment status,[55] race,[18, 19, 21, 27, 31, 60] and religion. [28] Moreover, being single was identified as another predictor of underuse for cancer screening. The results conclude that the lower income and less educated population with no health insurance coverage make lower use of cancer screening. Finally, this systematic review found that the place of residence also affected cancer screening participation. This information is very relevant since in LA a large number of people live far from urban regions or in rural-areas. Therefore, cancer screening needs to take this into account.

This literature review concluded that being covered by an insurance plan was a determinant for participating in breast and cervical cancer screening. In LA countries, health systems are often highly fragmented [61] and the population is covered by different configurations, often

involving a combination of private health insurance, social security and public health care systems. Access to healthcare, including screening services, varies across the population depending on the type of insurance people have.

The lack of reliable information on the characteristics of existing cancer screening programs across LA precludes any conclusions regarding a link between differences in healthcare systems and distribution of cancer screening underuse across income levels. Research, like that conducted previously in Europe by the International Agency for Research on Cancer (IARC), [62], would be essential to answer this question. Moreover, it is important to acknowledge that cancer screening by itself does not reduce cancer mortality and that it must come together with appropriate referral and timely treatment. Analysing the impact of socioeconomic status in accessing cancer treatment services would help identify other factors explaining the higher cancer mortality in populations with lower socioeconomic status.

This study has several limitations. The studies identified in the literature review were cross-sectional and therefore causation between the independent and dependent variables could not be established.[63] Cross-sectional studies may be affected by recall or social desirability bias in which the respondents might share inaccurate information.[64] For instance, in household surveys there is a trend to underreport income, due to forgetfulness,[65–68], among other reasons, and this could affect the validity of the income measure in the selected studies. Furthermore, most studies included in this review did not consider the economies of scale (meaning how the needs of a household grow with each additional individual), and only income was used to define wealth level.[69] The lack of information on the characteristics of cancer screening programmes in LA limited the capacity of this study to identify those screening programme characteristics linked to better performance in reducing social disparities. Although the possibility of publication bias exists, the nature of the studies included in this review makes publication bias unlikely: most studies were based on health surveys that are publicly available and could be easily reanalysed; each study included two or more socioeconomic variables that could have different impact on cancer screening access; and some of the included studies analysed data from the same survey obtaining similar results. Finally, this study did not analyse information about the impact of socioeconomic status in accessing other services in the continuum of cancer care such as HPV vaccination, cervical precancerous lesions, or invasive cancer. Getting this information in future studies would shed more light onto the factors that explain the higher cancer mortality in low socioeconomic strata.

However, this study has important strengths, providing evidence on the socioeconomic factors related to poor utilisation of cancer screening in LA. First, this study followed a rigorous process, collecting for the first time the available evidence on socioeconomic determinants that affect access to cancer screening in LA. This review used three databases relevant to the topic and another database (LILACS) specific to LA. We included studies in English, Portuguese and Spanish, thereby reducing the chances of losing relevant information. Moreover, the quality of the papers was assessed to ensure that the analysed papers were of acceptable quality.

## Conclusions and policy recommendations

This study suggests that income, education level, health insurance and marital status are all determinants for using cancer screening services in LA. The public health implication of this study is that it helped identify populations with lower use of cancer screening. The study also helped to identify research gaps that need to be addressed to fully understand cancer screening access in LA and contribute to the elaboration of relevant policies. The establishment of well-organized cancer screening strategies has helped some countries in LA such as Argentina [33] to reduce cancer disparities. However, a lot of work remains to be done. In order to reduce avoidable cancer deaths, we recommend that countries should:

1. Conduct research to understand the reasons behind the lower participation in cancer screening in populations of lower socioeconomic status.

2. Conduct research to explore inequalities in access to cancer screening in specific populations, such as people with disabilities or indigenous populations.

3. Develop effective information campaigns and culturally sensitive messages that proactively reach those populations that are under-users of cancer screening.

4. Develop population-based screening programs to recruit populations from different age groups and socioeconomic levels.

5. Upscale the introduction of approaches such as HPV self-sampling, mobile mammography and colonoscopy services to better reach socially vulnerable under-screened populations.

6. Make screening services accessible to uninsured populations and ensure that screening services are available at different times and days so that workers can access these services.

## Supporting information

**S1 File. PRISMA 2009 checklist.**
(DOC)

**S1 Table. Data extraction sheet headings.**
(DOCX)

**S2 Table. Modified NIH quality assessment tool for observational and cross-sectional studies.**
(DOCX)

## Author Contributions

**Conceptualization:** Bernardo Nuche-Berenguer, Dikaios Sakellariou.

**Data curation:** Bernardo Nuche-Berenguer.

**Formal analysis:** Bernardo Nuche-Berenguer, Dikaios Sakellariou.

**Investigation:** Bernardo Nuche-Berenguer.

**Methodology:** Bernardo Nuche-Berenguer, Dikaios Sakellariou.

**Project administration:** Bernardo Nuche-Berenguer.

**Resources:** Bernardo Nuche-Berenguer.

**Software:** Bernardo Nuche-Berenguer.

**Supervision:** Dikaios Sakellariou.

**Writing – original draft:** Bernardo Nuche-Berenguer, Dikaios Sakellariou.

**Writing – review & editing:** Bernardo Nuche-Berenguer, Dikaios Sakellariou.

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
