## [Decision Letter · Decision Letter 0]

12 Sep 2019

PONE-D-19-18831

Socioeconomic predictors for low use of cancer screening in Latin America: a systematic review

PLOS ONE

Dear Dr. Sakellariou,

Thank you for submitting your manuscript to PLOS ONE. After careful consideration, we feel that it has merit but does not fully meet PLOS ONE’s publication criteria as it currently stands. Therefore, we invite you to submit a revised version of the manuscript that addresses the points raised during the review process.

We would appreciate receiving your revised manuscript by Oct 27 2019 11:59PM. To enhance the reproducibility of your results, we recommend that if applicable you deposit your laboratory protocols in protocols.io, where a protocol can be assigned its own identifier (DOI) such that it can be cited independently in the future. For instructions see: http://journals.plos.org/plosone/s/submission-guidelines#loc-laboratory-protocols

We look forward to receiving your revised manuscript.

Kind regards,

Thomas Behrens

Academic Editor

PLOS ONE

2. In the Methods, please provide the start and end dates of the search timeframe (month and year). Furthermore, please clarify why the databases were searched from 2009 onwards, instead of from inception?

3. Please provide an analysis of potential publication bias in addition to the quality analysis that you have already performed.

Reviewers' comments:

Reviewer's Responses to Questions

**Comments to the Author**

1. Is the manuscript technically sound, and do the data support the conclusions?

Reviewer #1: Yes

Reviewer #2: Partly

2. Has the statistical analysis been performed appropriately and rigorously? 

Reviewer #1: Yes

Reviewer #2: N/A

3. Have the authors made all data underlying the findings in their manuscript fully available?

Reviewer #1: Yes

Reviewer #2: Yes

4. Is the manuscript presented in an intelligible fashion and written in standard English?

Reviewer #1: Yes

Reviewer #2: Yes

5. Review Comments to the Author

Reviewer #1: Bernardo Nuche-Berenguer and Dikaios Sakellariou present a well written manuscript; however some issues should be adequately addressed in order to improve the manuscript.

Method and extraction of this paper is not clear. Please add statements regarding:

-no. of publications in each language reviewed. In publications written in Spanish and Portuguese, were only abstract considered? or the authors extract the data from the body of the article? Add a flowchart describing the process of selection of publications in this section (Fig1).

Line 71- Among the socio-economic factors to search should be considered the distance of patients to urban areas, since in LA a large number of people live far from urban regions or in rural-areas?. As well as, data on ethnicity of patients I understand that unfortunately these data are not always available.

Line 89- Inclusion and exclusion criteria:

Authors should describe why the choice of these types of tumor (breast, cervical,

colorectal cancer) are most reported types of tumors in the region?

Line 94.- What is the reason to exclude the Caribbean islands?

Line 98- Data extraction:

Among the factors that can be included if the population lives in rural zones or located in cities?

Authors should describe if used a test statistic for OR? The confidence intervals should be placed in parentheses to separate them from the OR, in all cases.

Improve and use the same format for tables.

In the discussion, the authors mention that some LA countries have made progress in reducing cancer disparities, could provide concrete examples of the country and program, since depending on these recommendations can be extrapolated to other countries in the region. CicLAP and others grups?

Reviewer #2: The article is a narrative review of studies reporting on socioeconomic determinants for access to screening programs for three cancer sites between 2009 and 2018. Only 24 studies on the early detection of breast and cervical cancer were eligible for this analysis, but none for colorectal cancer. Therefore, the title could be changed to refer to screening for the early detection of breast and cervical cancer in women.

Abstract: The Result section should report on the effects of socioeconomic factors, separately for cervical and breast cancer.

Methods: The authors mention that information on the characteristics of the screening programs was frequently limited. However, the type of the program is pertinent for the analysis of socioeconomic factors. With regard to cancer (in particular cervical cancer) screening, it should be clarified, if the screening program was a research or pilot project with special aim to implement screening in a defined population such as a rural district (all socio-economic classes) or in special groups (indigenous women of low socio-economic status) or whether it is a general (nation-wide) free health-care program (for all socio-economic groups). Only the last type of screening programs allows an unbiased analysis of the participation by socio-economic status (SES), because all other screening programs are designed for selected subpopulations. If this information is limited, the analysis could be in two steps: for the studies selected and for those with sufficient information to be classified as a general population-based screening program.

Methods:

Table 1 indicates, with regard to cancer control, “Latin America” is a fractionated continent, but I have problems to understand this table aimed to show the search strategy. For example, the information on screening services and SES is lacking for several countries, such as Paraguay or Uruguay. A figure might be better. Most articles are from Brazil, and, for example, Argentina conducted an important pilot project to implement cervical cancer screening in a rural region. There was, for example, no report from Uruguay, classified with a “very high” Human Development Index and running a well-organized cancer registry.

Table 2 shows the search terms, this may be sufficient to delineate the search concept.

Results:

Because of differences in funding (higher for cervical than for breast cancer in Latin America) and in cancer risks by socio-economic status (SES) (higher risks for developing cervical cancer in women with low SES and higher risks for developing breast cancer in women with high SES), the analysis should be performed separately for these two cancer sites and presented in subtables.

Cervical cancer is of special concern in Latin America. Screening for the early detection of breast and colorectal cancer is less well organized. Cervical cancer screening is not just one step, but a stepwise procedure (HPV screening, if positive CIN screening). This should be considered when presenting the studies.

Challenges for cancer control in Latin America have been subject to comprehensive evaluations. Major problem is disparity in health care between metropolitan und rural regions. Therefore, the studies should be categorized by region (nation-wide, urban, rural).

A table showing the results by country and cancer site would be informative as first table of the Result section. The article should describe the characteristics of the region (country, nationwide or state or department), method (e.g. mammography) and target population more precisely in Table 3, separately for breast and cervical cancer. So far, results were presented in Table 3 in an “unstructured manner” with a variety of formatting errors and not informative due to the crude “listing” of studies. This table requires major revision and a harmonization of the different outcome variables, such as years of education or income, if possible, into low, medium, high SES. By the way, underreporting of the income is not just “forgetfulness”. This harmonized classification would allow a meta-analysis of the effect of SES by cancer site.

Discussion

There are excellent publications on cervical cancer in Latin America. Nation-wide HPV vaccination programs have been established in a many countries, for example also in Paraguay, with support of international agencies, including PAHO, covering all girls in schools. The next step is the detection of HPV infections and precancerous lesions (CIN) in young women. Here, SES may play a crucial role, but mainly with regard to the clinical treatment capacity of those tested positively with higher CIN stages, which is limited to metropolitan regions. Poorer women from rural regions cannot travel to distant health care centers. This problem is not captured when investigating the role of SES in CIN screening in rural regions.

6. PLOS authors have the option to publish the peer review history of their article (what does this mean?). If published, this will include your full peer review and any attached files.

Reviewer #1: Yes: Andrea Orue

Reviewer #2: No

---

## [Author Response · Author response to Decision Letter 0]

22 Oct 2019

Dear Editor,

Thank you very much for the thoughtful comments we received by the two reviewers and by the editorial office. We have revised the manuscript in response to these comments, as outlined below.

Sincerely,

The authors

Comments from the editorial office

RESPONSE: We have now addressed this.

2. In the Methods, please provide the start and end dates of the search timeframe (month and year). Furthermore, please clarify why the databases were searched from 2009 onwards, instead of from inception? 

RESPONSE: The search timeframe was January 2009 to July 2018. The reason why the databases were searched from 2009 is because we were looking for updated information as the implementation of new cancer norms and screening programs in Latin America is expected to have an impact in the access to cancer screening services. We have added this information in lines 95-98, ad in the last line of table 2

3. Please provide an analysis of potential publication bias in addition to the quality analysis that you have already performed. 

RESPONSE: Although a possibility of publication bias may exist we believe that the nature of the studies included in this review makes publication bias unlikely. First, most studies were based on health surveys that are publicly available and could be easily reanalyzed. Second, each study included two or more socioeconomic variables that could have different impact on cancer screening access. Third, some of the found studies were analyzing data from the same survey and obtained similar results. We have now added this information in lines 298-302.

4. Please include captions for your Supporting Information files at the end of your manuscript, and update any in-text citations to match accordingly. Please see our Supporting Information guidelines for more information:http://journals.plos.org/plosone/s/supporting-information. 

RESPONSE: We have now addressed this.

Reviewers' comments:

Reviewer #1: 

Bernardo Nuche-Berenguer and Dikaios Sakellariou present a well written manuscript; however some issues should be adequately addressed in order to improve the manuscript.

RESPONSE: Thank you for your contrastive feedback. Please find our detailed responses below

1. Method and extraction of this paper is not clear. Please add statements regarding:

-no. of publications in each language reviewed. 

RESPONSE: We added a clear statement about this in lines 111-113 and 132-142, and we also show this information in the PRISMA diagram. 127 articles were subjected to full review (74 in English, 36 in Portuguese and 17 in Spanish). Data were extracted from 24 articles (13 in English, 8 in Portuguese and 3 in Spanish.

2.In publications written in Spanish and Portuguese, were only abstract considered? or the authors extract the data from the body of the article? Add a flowchart describing the process of selection of publications in this section (Fig1). 

RESPONSE: Please see answer above. 

3. 

Line 71- Among the socio-economic factors to search should be considered the distance of patients to urban areas, since in LA a large number of people live far from urban regions or in rural-areas?. As well as, data on ethnicity of patients I understand that unfortunately these data are not always available.

RESPONSE: We revised the 24 selected articles to extract the specific information related to the impact of the place of residence on cancer screening participation. This information has been added to the column of independent variables in tables 4 and 5 and also a new subsection summarizing this information has been added in line 214-219 and also in the discussion (lines 268-271). The information about the impact of race ethnicity is briefly summarized in the subsection “Other factors”.

4. Line 89- Inclusion and exclusion criteria:

Authors should describe why the choice of these types of tumor (breast, cervical,

colorectal cancer) are most reported types of tumors in the region? 

RESPONSE: Breast, cervical and colorectal cancer are together with other cancer types such as prostate or lung cancer the most common cancer types in Latin America. Countries in Latin America have national screening programmes for breast, cervical and colorectal. While breast and cervical cancer screening are recommended by WHO, colorectal cancer screening is recommended by other organizations such as USPSTF or the American Cancer society. Analysis in the access to prostate cancer screening was excluded because although prostate cancer screening is available in some countries there are no widespread national screening programmes and the benefits of prostate cancer screening are not clear. This is now explained from lines 51-55 and again in lines 92-94 (inclusion and exclusion criteria).

5. Line 94.- What is the reason to exclude the Caribbean islands? 

RESPONSE: The Caribbean is a different sociopolitical region with a very different geographical and cultural characteristics. We feel that a similar analysis could be conducted in the Caribbean but that this should be done apart of the present analysis as the interpretation of the results may be different. This is explained in lines 99-101.

6. Line 98- Data extraction:

Among the factors that can be included if the population lives in rural zones or located in cities? 

RESPONSE: Thank for your comment. As stated above this information is now included in the results and discussion sections.

7.Authors should describe if used a test statistic for OR? The confidence intervals should be placed in parentheses to separate them from the OR, in all cases.

 RESPONSE: We made sure that all confidence internals (where available) were placed within parentheses. However, not all studies reported confidence intervals. The authors of the included studies used logistic regressions to calculate OR.

8. Improve and use the same format for tables. We streamlined the presentation of the tables and we also divided the main results table into two separate tables (tables 4 and 5) for clarity, as suggested by reviewer 2.

9. In the discussion, the authors mention that some LA countries have made progress in reducing cancer disparities, could provide concrete examples of the country and program, since depending on these recommendations can be extrapolated to other countries in the region. CicLAP and others grups? 

RESPONSE: We have added an example in lines 319-321.

Reviewer #2: 

10. The article is a narrative review of studies reporting on socioeconomic determinants for access to screening programs for three cancer sites between 2009 and 2018. 

RESPONSE: Thank you for your contrastive feedback. Please find our detailed responses below 

Only 24 studies on the early detection of breast and cervical cancer were eligible for this analysis, but none for colorectal cancer. Therefore, the title could be changed to refer to screening for the early detection of breast and cervical cancer in women.

RESPONSE: We believe that because the review set out to also explore colorectal cancer, the title should remain as it is. We highlighted in the findings the implication of the lack of research for colorectal cancer. 

11.Abstract: The Result section should report on the effects of socioeconomic factors, separately for cervical and breast cancer. 

RESPONSE: We added a sentence about the effects of socioeconomic factors in the abstract; as the effects are quite similar, we decided to mention together breast and cervical cancer screening in the abstract

12. Methods: The authors mention that information on the characteristics of the screening programs was frequently limited. However, the type of the program is pertinent for the analysis of socioeconomic factors. With regard to cancer (in particular cervical cancer) screening, it should be clarified, if the screening program was a research or pilot project with special aim to implement screening in a defined population such as a rural district (all socio-economic classes) or in special groups (indigenous women of low socio-economic status) or whether it is a general (nation-wide) free health-care program (for all socio-economic groups). Only the last type of screening programs allows an unbiased analysis of the participation by socio-economic status (SES), because all other screening programs are designed for selected subpopulations. If this information is limited, the analysis could be in two steps: for the studies selected and for those with sufficient information to be classified as a general population-based screening program. 

RESPONSE: Thank you for this interesting comment. The purpose of the study was to analyze the impact of socioeconomic factors in access to cancer screening. Most of the studies reporting this information were based on health surveys in which participants were asked about a number of socioeconomic aspects together with their past participation in cancer screening. Therefore, the information regarding the nature of the cancer screening is not available in these studies. We are aware that this is a limitation in the interpretation of the results of the study since the information of exposure to cancer screening is unknown. Although there are some global efforts to identify the type of cancer screening program available in each country, this information is currently unknown. 

Methods:

13. Table 1 indicates, with regard to cancer control, “Latin America” is a fractionated continent, but I have problems to understand this table aimed to show the search strategy. For example, the information on screening services and SES is lacking for several countries, such as Paraguay or Uruguay. A figure might be better. Most articles are from Brazil, and, for example, Argentina conducted an important pilot project to implement cervical cancer screening in a rural region. There was, for example, no report from Uruguay, classified with a “very high” Human Development Index and running a well-organized cancer registry. 

RESPONSE: Table 1 does not show the search strategy- this is shown is table 2. Our aim in Table 1 is to present the search concepts identified from the research question. The concepts presented in the three columns were all combined in the search, presented in table 2.

14. Table 2 shows the search terms, this may be sufficient to delineate the search concept. 

RESPONSE: While we agree that table 2 also shows the search concepts, these are more clearly presented in table 1, and linked to each of the terms in the questions, so we believe that to enhance clarity and transparency it would be good to keep both tables in the article.

15.Because of differences in funding (higher for cervical than for breast cancer in Latin America) and in cancer risks by socio-economic status (SES) (higher risks for developing cervical cancer in women with low SES and higher risks for developing breast cancer in women with high SES), the analysis should be performed separately for these two cancer sites and presented in subtables. 

RESPONSE: Thank you for this comment. We are now presenting the reslts in two separate tables, as suggested. 

16. Cervical cancer is of special concern in Latin America. Screening for the early detection of breast and colorectal cancer is less well organized. Cervical cancer screening is not just one step, but a stepwise procedure (HPV screening, if positive CIN screening). This should be considered when presenting the studies. 

RESPONSE: We agree that cancer screening by itself does not reduce cancer mortality and that it must come together with appropriate referral and timely treatment. However, the aim of this study was analyzing the impact of socioeconomic status in accessing cancer services and the gateway for cancer services should ideally be cancer screening. Getting information about the impact of socioeconomic status in accessing other services in the continuum of cancer care would be very interesting of course, however, the nature of the identified studies that were mostly survey-based does not allow to obtain this information. We added some explanation of this in lines 284-286 and 302-306.

17. Challenges for cancer control in Latin America have been subject to comprehensive evaluations. Major problem is disparity in health care between metropolitan und rural regions. Therefore, the studies should be categorized by region (nation-wide, urban, rural). 

RESPONSE: We revised the 24 selected articles to extract the specific information related to the impact of the place of residence on cancer screening participation. This information has been added to the column of independent variables in tables 4 and 5 and also a new subsection summarizing this information has been added in line 214-219 and also in the discussion (lines 268-271). 

18. A table showing the results by country and cancer site would be informative as first table of the Result section. The article should describe the characteristics of the region (country, nationwide or state or department), method (e.g. mammography) and target population more precisely in Table 3, separately for breast and cervical cancer. So far, results were presented in Table 3 in an “unstructured manner” with a variety of formatting errors and not informative due to the crude “listing” of studies. This table requires major revision and a harmonization of the different outcome variables, such as years of education or income, if possible, into low, medium, high SES. By the way, underreporting of the income is not just “forgetfulness”. This harmonized classification would allow a meta-analysis of the effect of SES by cancer site. 

RESPONSE: Thank you for this helpful comment. We thoroughly revised the main results table and split it into two (tables 4 and 5), which we think adds more clarity.

19. There are excellent publications on cervical cancer in Latin America. Nation-wide HPV vaccination programs have been established in a many countries, for example also in Paraguay, with support of international agencies, including PAHO, covering all girls in schools. The next step is the detection of HPV infections and precancerous lesions (CIN) in young women. Here, SES may play a crucial role, but mainly with regard to the clinical treatment capacity of those tested positively with higher CIN stages, which is limited to metropolitan regions. Poorer women from rural regions cannot travel to distant health care centers. This problem is not captured when investigating the role of SES in CIN screening in rural regions. RESPONSE: Thank you for this helpful comment. We added some explanation of this in lines 284-288 and 302-306.

---

## [Decision Letter · Decision Letter 1]

11 Nov 2019

Socioeconomic determinants of cancer screening utilisation in Latin America: a systematic review

PONE-D-19-18831R1

Dear Dr. Sakellariou,

We are pleased to inform you that your manuscript has been judged scientifically suitable for publication and will be formally accepted for publication once it complies with all outstanding technical requirements.

With kind regards,

Thomas Behrens

Academic Editor

PLOS ONE

Additional Editor Comments (optional):

Reviewers' comments:

Reviewer's Responses to Questions

**Comments to the Author**

1. If the authors have adequately addressed your comments raised in a previous round of review and you feel that this manuscript is now acceptable for publication, you may indicate that here to bypass the “Comments to the Author” section, enter your conflict of interest statement in the “Confidential to Editor” section, and submit your "Accept" recommendation.

Reviewer #1: All comments have been addressed

2. Is the manuscript technically sound, and do the data support the conclusions?

Reviewer #1: Yes

3. Has the statistical analysis been performed appropriately and rigorously? 

Reviewer #1: Yes

4. Have the authors made all data underlying the findings in their manuscript fully available?

Reviewer #1: Yes

5. Is the manuscript presented in an intelligible fashion and written in standard English?

Reviewer #1: Yes

6. Review Comments to the Author

Reviewer #1: All the observations were considered and the manuscript improved.

1. Method and extraction of this paper is not clear. Please add statements regarding:

-no. of publications in each language reviewed.

RESPONSE: We added a clear statement about this in lines 111-113 and 132-142, and we also show this information in the PRISMA diagram. 127 articles were subjected to full review (74 in English, 36 in Portuguese and 17 in Spanish). Data were extracted from 24 articles (13 in English, 8 in Portuguese and 3 in Spanish.

ok

2.In publications written in Spanish and Portuguese, were only abstract considered? or the authors extract the data from the body of the article? Add a flowchart describing the process of selection of publications in this section (Fig1).

RESPONSE: Please see answer above.

ok

3.Line 71- Among the socio-economic factors to search should be considered the distance of patients to urban areas, since in LA a large number of people live far from urban regions or in rural-areas?. As well as, data on ethnicity of patients I understand that unfortunately these data are not always available.

RESPONSE: We revised the 24 selected articles to extract the specific information related to the impact of the place of residence on cancer screening participation. This information has been added to the column of independent variables in tables 4 and 5 and also a new subsection summarizing this information has been added in line 214-219 and also in the discussion (lines 268-271). The information about the impact of race ethnicity is briefly summarized in the subsection “Other factors”.

4. Line 89- Inclusion and exclusion criteria:

Authors should describe why the choice of these types of tumor (breast, cervical,

colorectal cancer) are most reported types of tumors in the region?

RESPONSE: Breast, cervical and colorectal cancer are together with other cancer types such as prostate or lung cancer the most common cancer types in Latin America. Countries in Latin America have national screening programmes for breast, cervical and colorectal. While breast and cervical cancer screening are recommended by WHO, colorectal cancer screening is recommended by other organizations such as USPSTF or the American Cancer society. Analysis in the access to prostate cancer screening was excluded because although prostate cancer screening is available in some countries there are no widespread national screening programmes and the benefits of prostate cancer screening are not clear. This is now explained from lines 51-55 and again in lines 92-94 (inclusion and exclusion criteria).

5. Line 94.- What is the reason to exclude the Caribbean islands?

RESPONSE: The Caribbean is a different sociopolitical region with a very different geographical and cultural characteristics. We feel that a similar analysis could be conducted in the Caribbean but that this should be done apart of the present analysis as the interpretation of the results may be different. This is explained in lines 99-101.

ok

6. Line 98- Data extraction:

Among the factors that can be included if the population lives in rural zones or located in cities?

RESPONSE: Thank for your comment. As stated above this information is now included in the results and discussion sections.

perfect!

7.Authors should describe if used a test statistic for OR? The confidence intervals should be placed in parentheses to separate them from the OR, in all cases.

RESPONSE: We made sure that all confidence internals (where available) were placed within parentheses. However, not all studies reported confidence intervals. The authors of the included studies used logistic regressions to calculate OR.

ok

8. Improve and use the same format for tables. We streamlined the presentation of the tables and we also divided the main results table into two separate tables (tables 4 and 5) for clarity, as suggested by reviewer 2.

9. In the discussion, the authors mention that some LA countries have made progress in reducing cancer disparities, could provide concrete examples of the country and program, since depending on these recommendations can be extrapolated to other countries in the region. CicLAP and others grups?

RESPONSE: We have added an example in lines 319-321.

ok

7. PLOS authors have the option to publish the peer review history of their article (what does this mean?). If published, this will include your full peer review and any attached files.

Reviewer #1: No

---

## [Editor Report · Acceptance letter]

18 Nov 2019

PONE-D-19-18831R1 

Socioeconomic determinants of cancer screening utilisation in Latin America: a systematic review 

Dear Dr. Sakellariou:

I am pleased to inform you that your manuscript has been deemed suitable for publication in PLOS ONE. Congratulations! Your manuscript is now with our production department. 

With kind regards,

on behalf of

Prof. Thomas Behrens 

Academic Editor

PLOS ONE